# Minimization of the *Bacillus subtilis* divisome suggests FtsZ and SepF can form an active Z-ring, and reveals the amino acid transporter BraB as a new cell division influencing factor

Ilkay Celik Gulsoy[1☯], Terrens N. V. Saaki[2☯], Michaela Wenzel[2¤], Simon Syvertsson[1], Taku Morimoto[3], Tjalling K. Siersma[2], Leendert W. Hamoen[1,2*]

**1** Centre for Bacterial Cell Biology, Institute for Cell and Molecular Biosciences, Newcastle University, Newcastle, United Kingdom, **2** Bacterial Cell Biology and Physiology, Swammerdam Institute for Life Sciences, University of Amsterdam, Amsterdam, The Netherlands, **3** Graduate School of Information Science, Nara Institute of Science and Technology, Ikoma, Nara, Japan

☯ These authors contributed equally to this work.
¤ Current address: Division of Chemical Biology, Department of Biology and Biological Engineering, Chalmers University of Technology, Gothenburg, Sweden
* l.w.hamoen@uva.nl

## Abstract

Bacterial cytokinesis begins with polymerization of the tubulin homologue FtsZ into a ring-like structure at midcell, the Z-ring, which recruits the late cell division proteins that synthesize the division septum. Assembly of FtsZ is carefully regulated and supported by a dozen conserved cell division proteins. Generally, these proteins are not essential, but removing more than one is in many cases lethal. Therefore, it is still not fully clear how the different protein components contribute to cell division, and whether there is a minimal set of proteins that can execute cell division. In this study, we tried to find the minimal set of proteins that is required to establish an active Z-ring in the model bacterium *Bacillus subtilis*. By making use of known suppressor mutations we were able to find a gene deletion route that eventually enabled us the remove eight conserved cell division proteins: ZapA, MinC, MinJ, UgtP, ClpX, Noc, EzrA and FtsA. Only FtsZ and its membrane anchor SepF appeared to be required for Z-ring formation. Interestingly, SepF is also the FtsZ anchor in archaea, and both proteins date back to the Last Universal Common Ancestor (LUCA). Viability of the multiple deletion mutant was not greatly affected, although the frequency of cell division was considerably reduced. Whole genome sequencing suggested that the construction of this minimal divisome strain was also possible due to the accumulation of suppressor mutations. After extensive phenotypic testing of these mutations, we found an unexpected cell division regulation function for the branched chain amino acid transporter BraB, which may be related to a change in fatty acid composition. The implications of these findings for the role of SepF, and the construction of a minimal cell division machinery are discussed.

**Data availability statement:** All relevant data are in the manuscript and its supporting information files.

**Funding:** This work was supported by a Newcastle University (https://www.ncl.ac.uk/) Overseas Research Studentship, Horizon Europe Marie Sklodowska-Curie Actions (marie-sklodowska-curie-actions.ec.europa.eu/) (618452), and Nederlandse Organisatie voor Wetenschappelijk Onderzoek (https://www.nwo.nl/en) (12128) awarded to LWH. The funders had no role in study design, data collection and analysis, decision to publish, or preparation of the manuscript.

**Competing interests:** The authors have declared that no competing interests exist.

## Author summary

One of the key aims of synthetic biology is the construction of a minimal cell. Since bacteria are the simplest life forms, they are the preferred blueprint for such a cell. However, a functional synthetic cell needs to be able to divide, and we still do not know what set of proteins are minimally necessary for bacterial cell division. This also hampers a full mechanistic understanding of this crucial process. In this study we tried to find the minimal set of proteins required for cell division in the model bacterium *Bacillus subtilis*, following a specific stepwise gene deletion protocol. Eventually, we were able to remove 8 conserved cell division related proteins and found that for the first step of the assembly of the cell division machinery, only two proteins are necessary, SepF and FtsZ. This surprising finding provides new functional insights into the bacterial cell division process, a blueprint for a synthetic minimal cell division protein set, and in addition, the gene deletion process also revealed a previously unknown cell division regulation activity for the conserved amino acid transport protein BraB.

## Introduction

The design and construction of an autonomously growing minimal cell will provide unprecedented biotechnological opportunities, and importantly, would be the decisive proof that we fully understand the principles of life, albeit in its simplest form. A crucial step in this endeavor is to define what is minimally necessary for a cell to divide. Bacteria are the most simple autonomously replicating life forms and the intricate knowledge of their biological processes has been used to design the blueprint for the first synthetic cells [1,2]. A major step has been the construction of a minimal *Mycoplasma mycoides* cell containing only 473 genes [3]. However, this cell did not contain any known cell division gene, presumably because, as a mycoplasma, it lacks a cell wall and can propagate by spontaneous membrane extrusion, or blebbing [3,4]. For bacteria that contain a cell wall, it is still unclear what set of proteins is minimally necessary for cytokinesis. This hampers a full understanding of the cell division process. Therefore, we examined whether there is a minimal set of proteins that can execute cell division in the Gram-positive model bacterium *Bacillus subtilis*.

Generally, bacterial cell division begins with polymerization of the conserved protein FtsZ into a ring-like structure at midcell. This so-called Z-ring functions as the scaffold for all the other cell division proteins, and together this complex is also referred to as the divisome. Almost all bacteria and many archaea contain FtsZ. Cell wall-lacking mycoplasma, and protoplasts of *B. subtilis* and *Escherichia coli* that can grow without a cell wall, so-called L-form cells, do not require FtsZ for propagation [3–6]. However, for cells that do contain a cell wall, a divisome machinery must be in place to synthesize the septum, which requires the presence of FtsZ. FtsZ is a tubulin homologue, and requires GTP hydrolysis for polymerization [7–10]. Polymerization of FtsZ at midcell is carefully regulated by several conserved cell division proteins, as schematically depicted for the *B. subtilis* situation in Fig 1. Firstly, FtsZ polymers need to be attached to the cell membrane, which in *B. subtilis* is accomplished by two different proteins, FtsA, present in most Gram-negative and Gram-positive bacteria, and SepF, which is conserved in Gram-positive bacteria, cyanobacteria and certain archaea [11–16]. Both proteins directly interact with FtsZ and both contain an amphipathic helix that functions as a membrane anchor [11,14,17]. Another conserved FtsZ-binding protein, EzrA, contains a transmembrane domain and is assumed to form large arch-like structures encompassing FtsZ polymers [18,19]. EzrA does not function as a membrane anchor for the Z-ring, but instead is

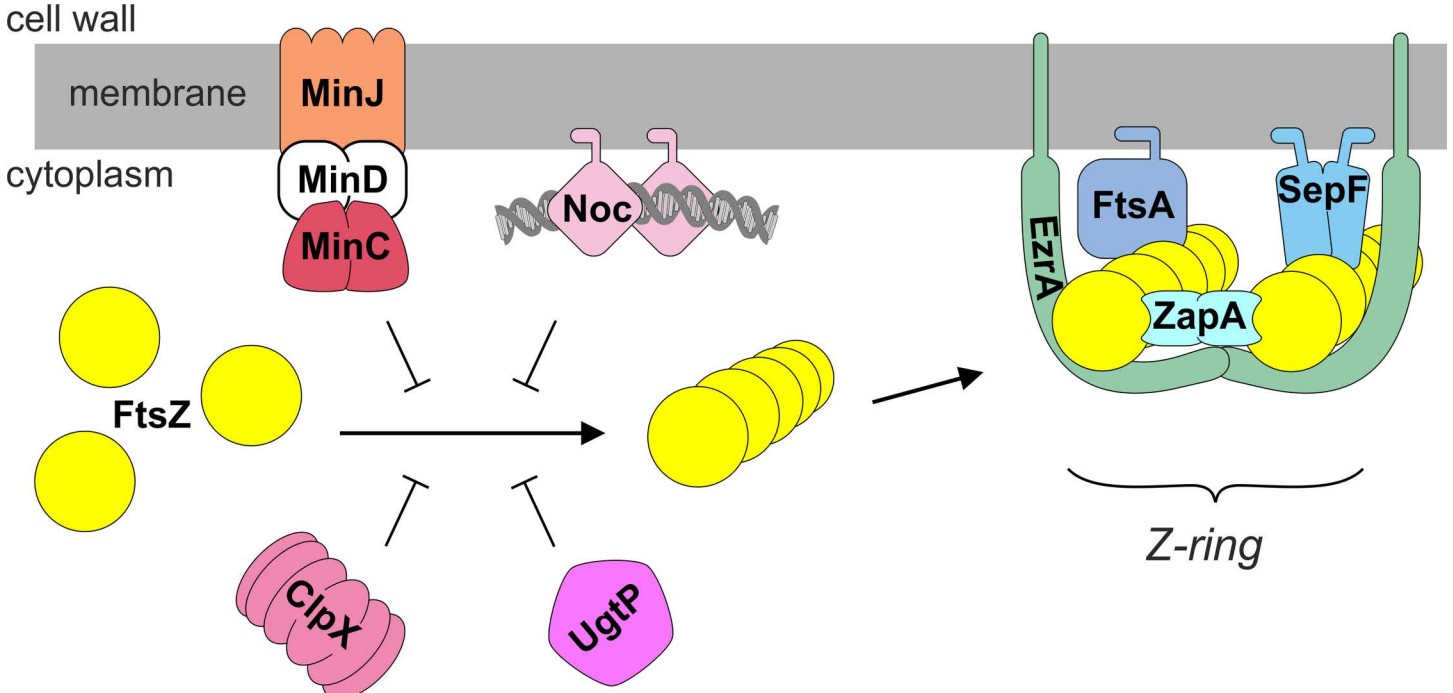

**Fig 1. Regulation and assembly of the Z-ring in *B. subtilis*.** FtsZ (yellow) monomers form polymers that bundle together to form a Z-ring. Both FtsA and SepF anchor FtsZ to the cell membrane. ZapA promotes bundling of FtsZ polymers, whereas EzrA appears to stabilize and regulate this complex by forming arcs. MinC inhibits FtsZ polymerization close to nascent cell division sites and cell poles, and is tethered to the membrane by MinD, which itself is recruited to the cell pole and division sites by MinJ. Nucleoid occlusion is regulated by Noc that binds both to the chromosome and the cell membrane. The protein chaperone ClpX stimulates recycling of FtsZ monomers after polymerization, and UgtP functions as a metabolic sensor that inhibits FtsZ polymerization depending on the levels of UDP-glucose. See main text for more details. The figure was made using CorelDRAW software.

required for the recruitment of the glycosyltransferase-transpeptidase PBP1 from the lateral wall to the division site, and can inhibit FtsZ polymerization [20,21]. The conserved protein ZapA forms bridges between FtsZ polymers and stabilizes the Z-ring [22,23]. After completion of the Z-ring, the so-called late cell division proteins arrive. These conserved transmembrane proteins are responsible for synthesis of the septal wall that divides the mother cell into two daughter cells, and comprise the peptidoglycan-transglycosylase FtsW and its cognate transpeptidase Pbp2B [24,25]. Assembly of these proteins at the Z-ring requires the bitopic membrane proteins FtsL, DivIB and DivIC, which form a stable tripartite complex. It is assumed that this complex regulates the recruitment of the late cell division proteins to the Z ring [26].

In Gram-positive bacteria it is unclear how the Z-ring is attached to the late cell division proteins. It is unlikely that FtsZ itself interacts with the late proteins, since the FtsZ polymers are separated from the cell membrane by the interaction with its membrane anchor FtsA and SepF. Cryo-EM studies with the Gram-negative bacteria *Caulobacter crescentus* and *E. coli* have shown that this distance is approximately 15–16 nm [27,28]. Moreover, the late cell division transmembrane proteins lack large cytoplasmic domains. Extensive mutagenesis studies with *E. coli* suggests that in Gram-negative bacteria FtsA forms the main link with the late cell division proteins, via interaction with the transmembrane cell division protein FtsN [29]. Possibly, FtsA plays a comparable role in *B. subtilis*, however, this bacterium can still grow and divide when FtsA is absent and also lacks a FtsN homolog [12,13]. Apparently, other Z-ring proteins can recruit the late cell division proteins in this organism.

Formation of the Z-ring needs to be carefully regulated since both the timing and position of division has to be coordinated with cell growth and chromosome replication. In many rod-shaped bacteria the conserved protein pair MinCD accumulates at cell poles and nascent division septa, and prevents non-functional divisions in these areas. MinC forms a complex with the membrane associated protein MinD and interacts directly with FtsZ, thereby inhibiting polymerization [30–33]. In *B. subtilis*, the MinCD complex is recruited to cell poles and nascent division sites by the transmembrane protein MinJ [34,35]. It has been postulated that MinJ is also required for efficient disassembly of the divisome after cell division has been completed [36]. Nucleoid occlusion prevents division through newly formed daughter chromosomes, and in *B. subtilis* this is executed by Noc, which binds DNA and is attached to the cell membrane by means of an amphipathic helix [37,38]. It is assumed that the resulting membrane-associated nucleoprotein complex physically interferes with efficient polymerization of FtsZ close to the cell membrane [38]. Cell division must also be coordinated with nutrient availability. In *B. subtilis* this is established by the glucosyltransferase UgtP which uses UDP-glucose as substrate for the synthesis of the cell wall glycolipid lipoteichoic acids, but also binds to and inhibits FtsZ [39]. Finally, FtsZ forms dynamic polymers by means of treadmilling [40], and the presence of a sufficient pool of unassembled FtsZ is partially assured by the activity of the conserved protein chaperone ClpX, which directly binds to FtsZ [41].

All these conserved cell division related proteins, except for FtsZ, can be individually inactivated in *B. subtilis* without blocking cell division. However, removing more than one of them generally results in very sick or lethal phenotypes, such as the combined inactivation of either SepF and FtsA, SepF and EzrA, EzrA and ZapA, EzrA and MinC, EzrA and Noc, MinC and Noc or MinC and ClpX [12,13,18,22,42]. This suggests that maybe almost all of these proteins are required to establish division, but how this mechanistically would work is conceptually difficult to understand due to the diversity in activities of these proteins. Unfortunately, the synthetic lethality makes it difficult to determine whether there is a minimal set of proteins that can support the formation of a functional Z-ring, thus hampering a full understanding of how this scaffold works. In an attempt to determine the minimal set of cell division proteins, we tried a deletion strategy whereby negative regulators were first removed, in the hope that this would suppress lethal combinations. Surprisingly, after several dead-ends, we were able to obtain a strain that lacked all cell division related proteins shown in Fig 1 except for SepF and FtsZ. This indirectly suggests that SepF can recruit late cell division proteins, which is interesting considering that SepF is conserved in bacteria and archaea that do not contain FtsA. However, whole genome sequencing revealed that this remarkable feat was, very likely, made possible due to the accumulation of suppressor mutations, of which several map to known cell division and cell wall synthesis genes. Interestingly, we also found an unexpected suppressor mutation that led to the discovery of a new cell division regulator function for the branched-chain amino acids transporter BraB. Inactivation of this gene appeared to change the lipid composition of cells. Implications of these findings for the assembly of the divisome, and the construction of a minimal cell division machinery, are discussed.

## Results

### Systematic removal of Z-ring modulating proteins

To define the minimal set of proteins required for cell division in *B. subtilis*, we set out to remove as many FtsZ-interacting proteins as possible. The natural genetic transformation ability of *B. subtilis* facilitates the combination of multiple genomic mutations, however the number of antibiotic resistance markers necessary for selection are limited and therefore we made use of an established marker-less gene deletion method [43]. In short, an

IPTG-inducible *mazF* toxin cassette was integrated next to a cell division gene, and subsequent induction of the MazF toxin resulted in the excision of the toxin cassette together with the cell division gene due to intra-molecular homologous recombination. The *mazF* toxin cassettes were separately introduced into the 168 wild type strain, and chromosomal DNA from these single insertion strains was used to construct strains with multiple deletions using natural genetic transformation. This procedure was chosen since chromosomal DNA integrates with much higher efficiencies compared to plasmid DNA. This strategy can be important as cell division defects can affect natural transformation. Deletion strains were labelled BMD (Bacillus Minimal Divisome) and they were grown in LB medium enriched with 1% glucose and 10 mM Mg$^{2+}$. The reason for these additions was that some deletion strains grew slower in LB medium, which could be mitigated by adding glucose. Extra Mg$^{2+}$ (10 mM) was added because of the *ugtP* deletion that we wanted to include (Fig 1) [44]. UgtP is required for synthesis of lipoteichoic acids, and the absence of this cell wall component results in bulging, which can be prevented by the addition of extra Mg$^{2+}$ in the medium [44].

We have tried different combinations of successive gene deletions, and some were more successful than others (Fig A in S1 Appendix). Eventually, we succeeded in deleting 8 cell division genes, as shown in Fig 2A. We started out with a Δ*zapA* deletion mutant (BMD1), since this mutant by itself has no clear phenotype [22]. Next, we removed the genes coding for the negative regulators MinC and UgtP (BMD2 and BMD3, respectively). This enabled us to remove the genes *minJ* and *ezrA* (BMD5 and BMD6, respectively). These multiple deletions had only a mild effect on either growth or cell length (Fig 2B and 2C). This in itself was surprising since it has been reported that a Δ*zapA* Δ*ezrA* double deletion grows very slowly and is filamentous [22]. Inactivation of the negative regulator ClpX was more complicated since this protein is also required for the development of genetic competence in *B. subtilis*. However, this requirement can be bypassed by removing the pleiotropic transcriptional regulator Spx [45]. Therefore, *spx* was deleted prior to *clpX* (BMD7 and BMD9, respectively). When we subsequently tried to delete *noc* (BMD12), only a few transformants were obtained, and a PCR check revealed that, although *noc* was absent in these transformants, the *ezrA* gene was restored due to a second double crossover event, which was possible since we used chromosomal DNA from a strain that contains only a *noc* deletion. The repair of *ezrA* was in a way not surprising since it has been shown that a Δ*noc* Δ*ezrA* double mutant is very sick [42]. Further attempts to delete one of the remaining cell division genes (*ezrA*, *sepF*, *ftsA*) using the marker-less method were unsuccessful. To be sure that it was indeed not possible to delete these genes, we tried to remove them by using chromosomal DNA from strains in which these genes were replaced by an antibiotic resistance marker. Interestingly, using this approach, we were able to remove subsequently *ezrA* and *ftsA*, resulting in BMD14 and BMD27, respectively. Deletion of these genes slightly reduces the growth rate and resulted in very long cells, comparable to a single *ftsA* mutant (Fig 2B and 2C). BMD27 lacked MinJ and all the main FtsZ-interacting proteins shown in Fig 1, except for SepF. Despite several attempts, we were unable to delete *sepF* in either BMD12 or BMD14.

## Phenotype of BMD27

To investigate whether BMD27 makes Z-rings, an ectopic xylose-inducible *ftsZ-gfp* reporter fusion was introduced (strain TNVS385). Fig 3A shows microscopic images of a representative cell expressing the FtsZ reporter fusion. The absence of 8 cell division proteins resulted in long filamentous cells with less Z-rings and polar deformation that seem to be caused by aberrant polar divisions (Fig 3B). It should be mentioned that in the stationary phase these cells became much shorter as they were apparently able to complete delayed division (Fig E in S1 Appendix). Except for the multiple polar minicells, these long cells strongly resemble an

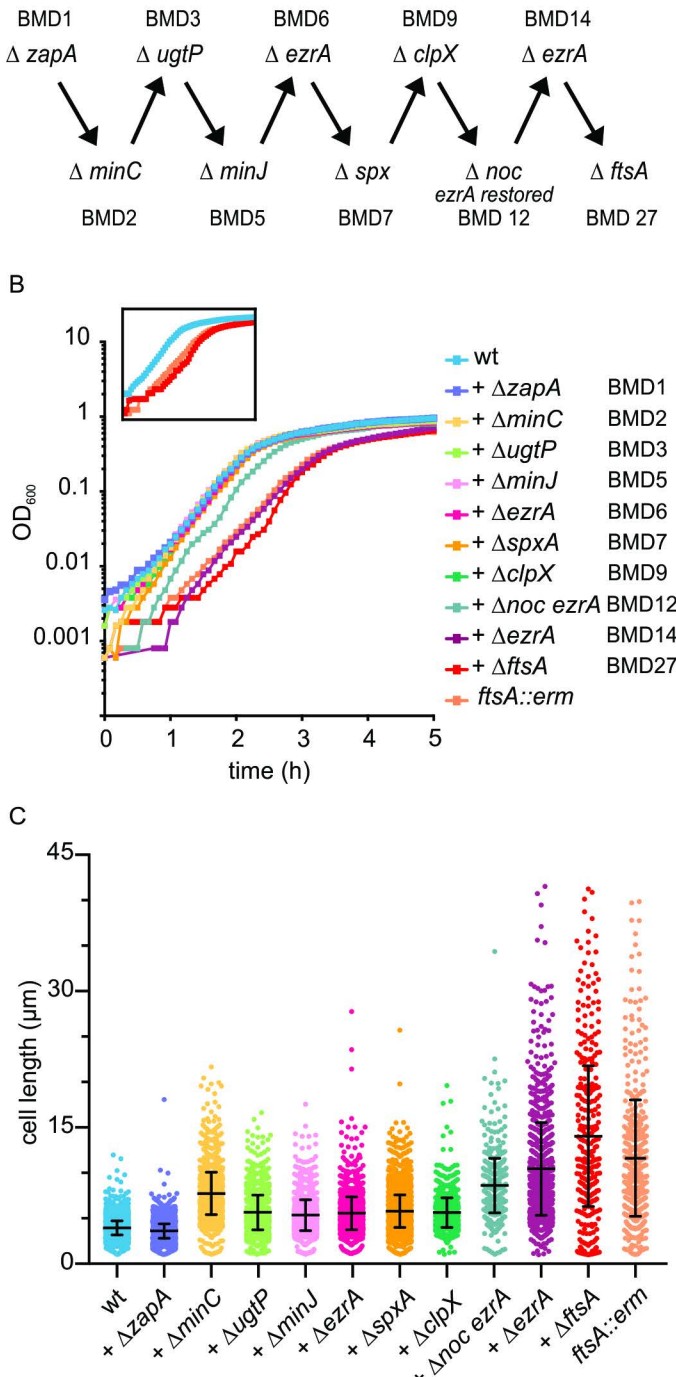

**Fig 2. Stepwise deletion of 8 cell division genes.** (A) Schematic overview of the sequential deletion of 8 conserved cell division proteins in *B. subtilis*. Of note, *spx* is not related to cell division, however a *spx* deletion restores genetic competence in a Δ*clpX* mutant. The resulting mutant strains were labelled BMD. See main text for more details. (B) Growth curves of BMD strains grown in microtiter plates at 37°C in LB medium supplemented with 1% glucose and 10 mM MgSO$_4$. Insert highlights the growth curves of wild type, BMD27, and the single *ftsA::erm* deletion strains. Graphs are representatives of at least 3 biological experiments, the 2 others are shown in Fig B in S1 Appendix. (C) Cell length measurement of the BMD strains. 3 biological replicates, for each more than 400 cells measured. Mini cells were not included. Mean and standard deviations are indicated.

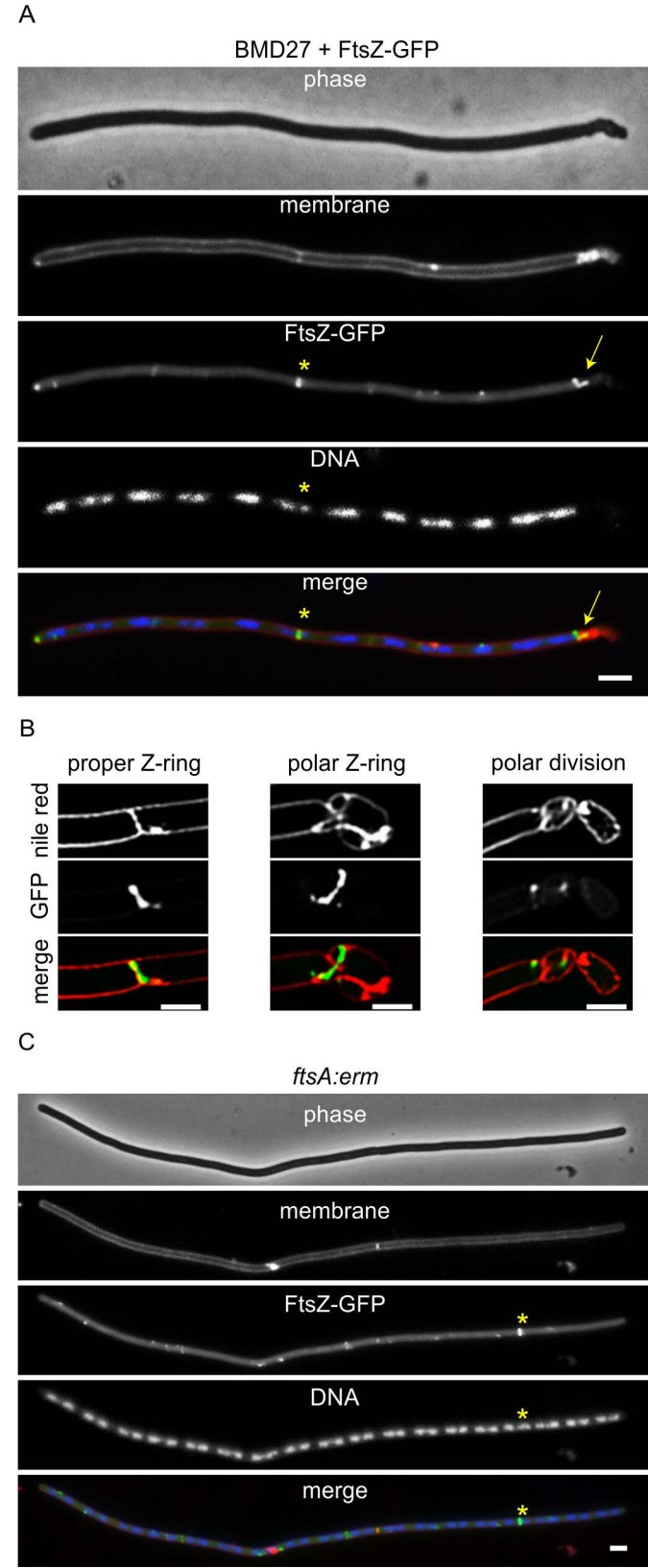

**Fig 3. Phenotype of BMD27.** (A) Fluorescence microscopy image of BMD27 cells expressing an ectopic FtsZ-GFP reporter fusion (green) (strains TNVS385). Membrane (red) and nucleoids (blue) were stained with FM5-95 and

DAPI, respectively. FtsZ-GFP forms normal Z-rings (asterisk) and aberrant helical structures at cell poles. The Z-ring in this example seem to bisect a nucleoid. For a large field image of several cells see Fig C in S1 Appendix. (B) Examples of SIM images of BMD27 cell poles containing several minicells and aberrant Z-structures. (C) Fluorescence microscopy image of an *ftsA*::*erm* mutant expressing an ectopic FtsZ-GFP reporter fusion (strain TNVS553). The star marked Z-ring forms over a nucleoid. For a large field image of several cells see Fig D in S1 Appendix. Cells were grown to OD$_{600}$ ~ 0.5 at 30 °C in LB supplemented with 10 mM MgSO$_4$, 1% fructose, and 1% xylose to induce FtsZ-GFP. Fructose was used instead of glucose because of glucose-dependent catabolite repression of the xylose promoter [105]. Scale bars, 2 μm.

*ftsA* single deletion mutant (Fig 3C, strain TNVS553). In 39% of the TNVS385 cells FtsZ formed aberrant helical structures and long arcs (Fig 3A). In the *ftsA*::*erm* single deletion mutant this was seen for approximately 21% of the Z-structures. The Z-ring in Fig 3A seems to bisect a nucleoid. This occurred regularly, with a frequency twice that of the *ftsA*::*erm* single deletion mutant (~43% and ~22%, respectively). Nevertheless, more than half of the Z-rings assembled in between nucleoids, indicating that some form of nucleoid occlusion is still functional in the multiple deletions strain TNVS385.

Previous studies have reported that the Min and nucleoid occlusion systems are not required for the positioning of Z-rings in the middle of cells in between the newly formed daughter chromosomes in *B. subtilis* and *E. coli* [46,47]. As shown in Fig 4, we found the same, and even in the multiple deletion strain BMD14 with an average cell length of more than 10 μm, there was still some preference for division at midcell (Fig 4). However, removing FtsA seems to abolish midcell selection, which was also the case for a *ftsA* single deletion mutant (Fig 4). Since these mutants forms cells with a comparable length as the BMD14 mutant containing FtsA (Fig 2C), it seems like FtsA has some regulatory role in FtsZ ring positioning.

## Whole genome sequencing

Since we were able to delete gene combinations that normally would have resulted in lethal phenotypes, it might be that suppressor mutations were collected during the process. To examine this, whole genome sequencing of all 10 deletion strains was performed. In total 3 silent mutations, 9 substitutions, 2 nucleotide insertions, and 8 deletions were found (Table 1), and their positions on the genome are schematically presented in Fig 5.

The first mutations occurred when *minC* was deleted (BMD2), yielding a point mutation (M66R) in the proton/glutamate symporter GltP [48]. The subsequent deletion of *ugtP* (BMD3) did not result in any mutation, however the removal of *minJ* (BMD5) caused the deletion of the complete 22 kb long Integrative and Conjugative Element ICEBs1. It has been shown that the inactivation of *ugtP* increases ICEBs1 conjugation efficiency [49], which may have facilitated the excision of ICEBs1. Subsequent deletion of *ezrA* (BMD6) resulted in a 6-basepair deletion in the N-terminal domain of the membrane protease protein DdcP [50], which has recently been shown to degrade YneA, the inhibitor of cell division during the SOS response [51]. How this would suppress any cell division phenotype is unclear. Deletion of *spx* did not cause any spontaneous mutations, however, the subsequent deletion of *clpX* (BMD9) yielded a 1.6 kb long deletion containing three genes, *mgsR, yqgY* and *yqgX*, the latter two coding for unknown proteins. MgsR moderates the expression of a subset of SigB regulated stress genes and is controlled by proteolytic degradation by the ClpXP protease complex [52,53]. The increased stability of MgsR after removal of ClpX might have provided selection pressure favoring this 1.6 kb genomic deletion. In addition, BMD9 contains point mutations in *uxuA* and *yoaN*, coding for a mannonate dehydrogenase and an inner spore coat protein, respectively [54,55].

Subsequent deletion of *noc* (BMD12) restored the *ezrA* gene, due to an additional double crossover event, and in addition to this, the strain acquired several spontaneous mutations

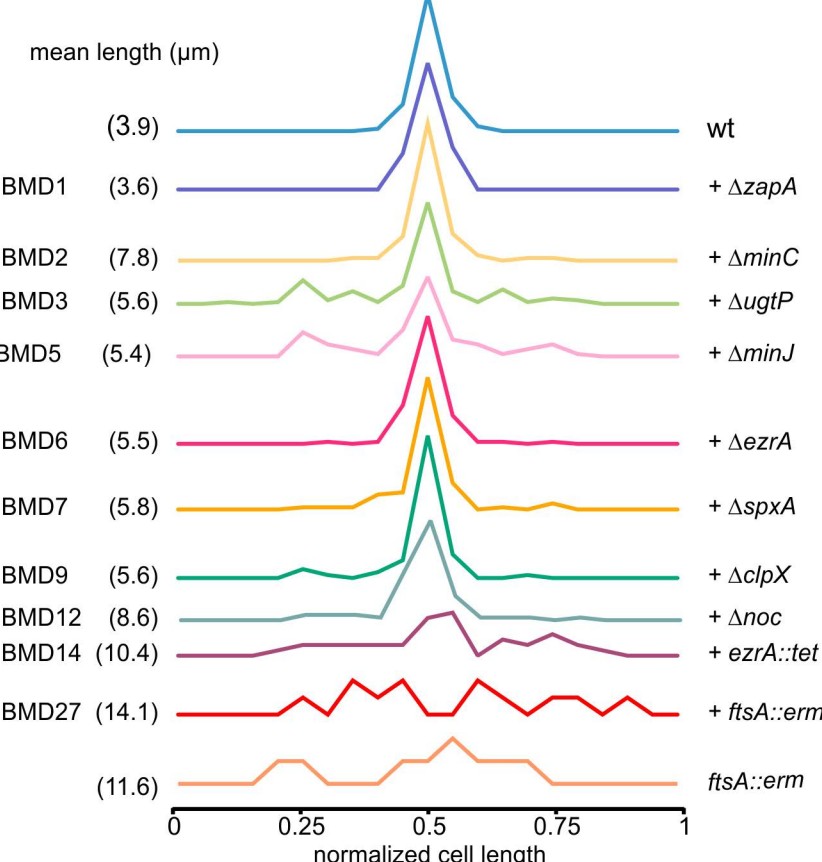

**Fig 4. Position of nascent cell division sites in deletion mutants.** The relative position of division sites was determined by staining cells with the fluorescent membrane dye FM5-95. The distance to one cell pole was measured and divided by the total cell length. To better indicate midcell preference, division positions were randomized relative to the left and right cell poles. Mean cell lengths are indicated between brackets. Data based on 3 biological replicates, minimally 75 cells were measured for the wild type and BMD1-12 strains, and minimally 50 cells for BMD14 and the *ftsA::erm* strain. Cells were grown in LB supplemented with 10 mM MgSO$_4$ and 1% glucose at 37 °C to exponential phase.

among which in *ponA* (G145R) and *spoVG* (E90*). Interestingly, the glycosyltransferase-transpeptidase PBP1, encoded by *ponA* and involved in lateral and septal peptidoglycan synthesis, is normally recruited to the cell division site by EzrA [20]. The conserved RNA binding protein SpoVG is a negative regulator of asymmetric cell division during sporulation [56]. In addition to these 2 mutations, BMD12 also obtained a triple amino acid deletion in the proline permease PutP (aa 356–358), and amino acid substitutions in the hypothetical proteins YuiF (L64F) and YwoF (G333V), the membrane protease SppA (A96V) [57], and a stop codon in the hypothetical protein YpmT (D31*).

When *ezrA* was again deleted in strain BMD14, mutations in *braB* (S343P), *rapA* (F186L) and *sftA* (P375S) were found. BraB is a branched-chain amino acid transporter [58], and RapA is a phosphatase involved in the quorum sensing regulation of sporulation [59]. The mutation in *sftA* is an interesting one since SftA is a conserved membrane-attached DNA translocase, with some homology to FtsK in *E. coli*. The protein binds to the Z-ring supported by both SepF and FtsA [60]. It has been speculated that the protein, like FtsK, prevents trapping of chromosomal DNA during septum synthesis by pumping DNA away

**Table 1. List of spontaneous mutations in the genomes of the different BMD mutants. Multi-nucleotide and gene deletions are indicated by Δ, and a stop codon mutation by \*.**

| deletion step | location | type | function |
|---|---|---|---|
| Δ*zapA* (BMD1) | – | – | – |
| Δ*minC* (BMD2) | *gltP* | M66R | H/glutamate symporter |
|  | intergenic | ATTTA > ATTA | between stop codons of *ywbE* and *ywbD* |
| Δ*ugtP* (BMD3) | – | – | – |
| Δ*minJ* (BMD5) | ICEBs1 | Δ 22 kb | integrative and conjugative element ICEBs1 |
|  | *yesO* | P305P | uptake of rhamnose oligosaccharides |
| Δ*ezrA* (BMD6) | *sigB* | G52G | general stress response sigma factor |
|  | *ddcP* | Δ I15-L16 | DNA damage checkpoint recovery protease, targets YneA |
|  | *htpG* | S336S | chaperone protein, heat shock response |
| Δ*spx* (BMD7) | – | – | – |
| Δ*clpX* (BMD9) | *uxuA* | P211L | mannonate dehydrogenase |
|  | *yoaN* | L140H | oxalate decarboxylase, spore coat protein |
|  | *mgsR* | Δ *mgsR-yqgX* | control of subset of SigB-regulated stress genes |
|  | *yqgY* | Δ *mgsR-yqgX* | unknown function |
|  | *yqgX* | Δ *mgsR-yqgX* | unknown function |
| Δ*noc* (BMD12) | *putP* | Δ L356-V358 | proline permease |
|  | *ypmT* | D31\* | unknown function |
|  | *ponA* | G143R | penicillin-binding protein PBP1 |
|  | *spoVG* | E90\* | control of asymmetric cell division (fore spore) |
|  | *yuiF* | L64F | unknown function |
|  | *ywoF* | G333V | unknown function |
|  | *sppA* | A96V | signal peptide peptidase |
|  | *ezrA* | restored |  |
| Δ*ezrA* (BMD14) | *braB* | S343P | branched-chain amino acid transporter |
|  | *sftA* | P375S | DNA translocase (chromosome dimer resolution) |
|  | *rapA* | F186L | quorum sensing regulation of Spo0A activity |
| Δ*ftsA* (BMD27) | *rsgI* | Δ *rsgI-ogt* | control of SigI activity (control of heat shock genes) |
|  | *sspD* | Δ *rsgI-ogt* | small acid-soluble spore protein |
|  | *ykrK* | Δ *rsgI-ogt* | repressor of HtpX |
|  | *htpX* | Δ *rsgI-ogt* | stress-responsive membrane protease |
|  | *ktrD* | Δ *rsgI-ogt* | low affinity potassium transporter |
|  | *ykzP* | Δ *rsgI-ogt* | unknown function |
|  | *ykzE* | Δ *rsgI-ogt* | unknown function |
|  | *ykrP* | Δ *rsgI-ogt* | unknown function |
|  | *kinE* | Δ *rsgI-ogt* | two-component sensor kinase, Spo0A activation |
|  | *ogt* | Δ *rsgI-ogt* | methylguanine DNA alkyltransferase (DNA-alkylation repair) |

from a closing septum. However, it is more likely that this function is executed by the FtsK homologue SpoIIIE in *B. subtilis* [60–63]. In *E. coli*, FtsK is an essential component of the divisome, but in *B. subtilis*, SftA is not required for cell division, although inactivation of the protein leads to slightly elongated cells [61,62,64]. It was shown that overexpression of SftA blocks cell division in an *ezrA* mutant, possibly because of interference with the assembly of SepF [60]. This activity may account for the spontaneous mutation in SftA. The disappearance of the *gltP*, *ezrA* and *sppA* mutations in BMD3, BMD12 and BMD14, respectively, can be explained by an extra double crossover event, as explained earlier. However, we have no

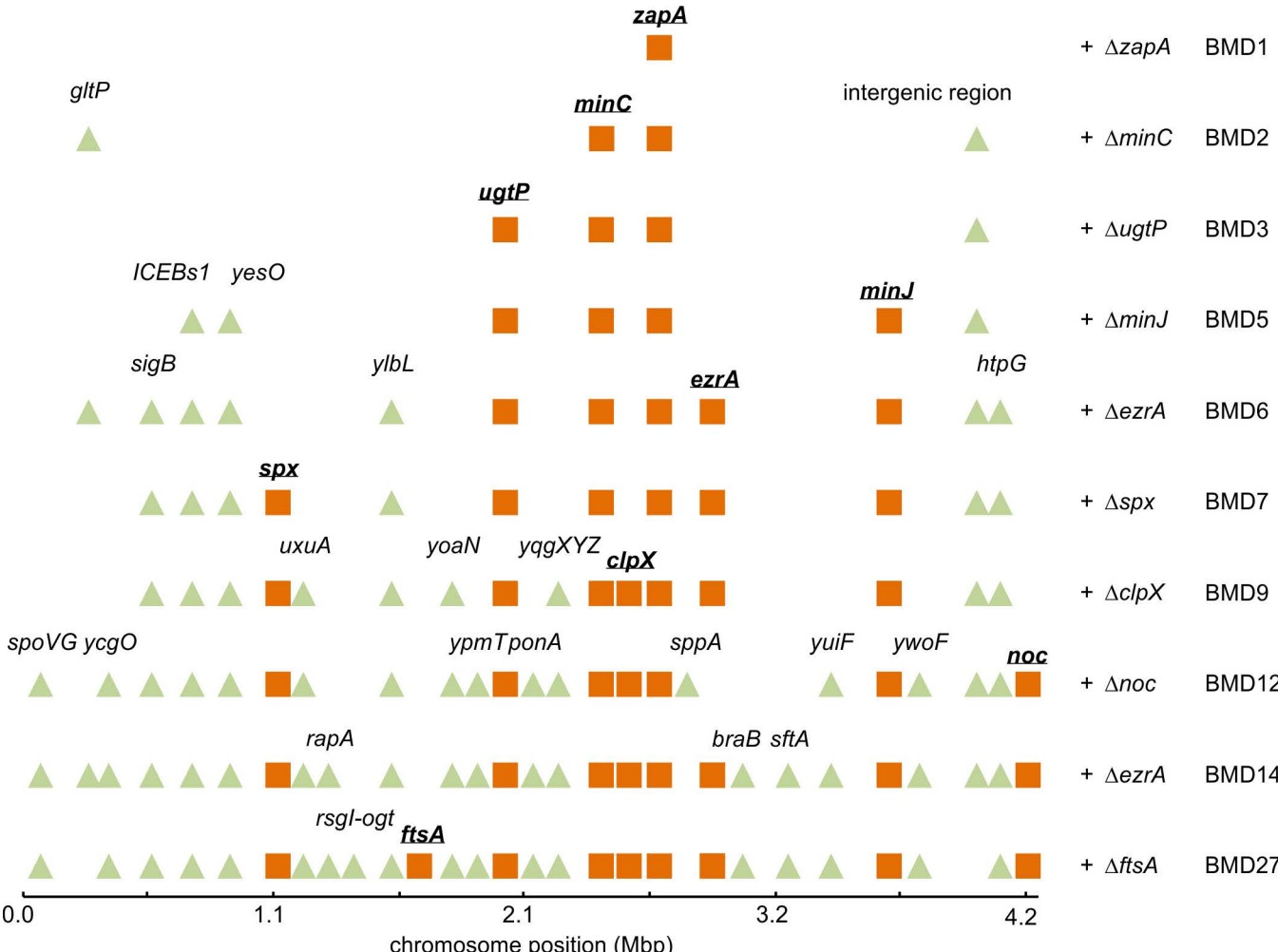

**Fig 5. Location of spontaneous mutations in the BMD genomes.** Schematic depiction of spontaneous mutations identified by whole genome sequencing on the 4.25 Mbp *B. subtilis* genome (X-axis). Deleted cell division genes are indicated as orange squares, and spontaneous mutations are indicated as green triangles. The *rsgI-ogt* deletion covers 10 genes. See main text for details.

explanation why the 2 bp deletion (GCCCA to GCC) in *gltP* in BMD 2 reappeared in BMD6 and again in BMD14.

Ultimately, removing *ftsA* (BMD27) yielded a large 6-kb deletion, covering 10 genes: *rsgI*, *sspD*, *ykrK*, *htpX*, *ktrD*, *ykzP*, *ykzE*, *ykrP*, *kinE* and *ogt*. RsgI is an anti-sigma factor controlling the activity of sigma I, which is required for the activation of general stress proteins, but also stimulates expression of the actin-homologue and cell shape-determining protein MreBH, and the undecaprenyl pyrophosphate phosphatase BcrC, involved in synthesis of the carrier lipid for cell wall synthesis [65]. Inactivation of RsgI has been shown to suppress certain cell shape defects in *B. subtilis* [66]. SspD is a protein found in spores [67], and KinE is a two-component sensor kinase involved in sporulation [68]. HtpX is a stress-responsive membrane protease whose expression is controlled by YkrK [69], and *ktrD* codes for a potassium transporter [70]. Lastly, the genes *ykzP*, *ykzE* and *ykrP* code for proteins with unknown functions, and the *ogt* gene codes for O6-methylguanine DNA alkyltransferase required for DNA-alkylation repair [71].

## Suppressor mutations

To test whether the suppressor mutations might play a role in cell division, we constructed deletion strains of all mutated genes in the wild type background, except for those that were located in the ICEBs conjugation element, and measured the effect on cell length (Fig 6A). The most apparent results were the longer cells of the *ponA* and *sftA* mutants, which deletions are known to increase cell length [61,62,72], and the shorter cell length of the *braB* mutant.

Mutations that affect cell division are more likely to show a clear phenotype when cell division is disturbed. This can, e.g., be achieved by treating cells with the FtsZ inhibitor 3-methoxybenzamide (3-MBA) [73]. To test whether the different deletion mutants showed any resistance to 3-MBA, the mutants were spotted onto agar plates containing increasing concentrations of 3-MBA. As shown in Fig 6B, the *ponA* and *braB* deletion mutants were clearly more resistant to 3-MBA, indicating that these are likely functional suppressor mutations in the BMD strains.

The efficacy of 3-MBA might be affected by changes in the cell envelope. Therefore, we assessed the results by testing the sensitivity of the most relevant deletion mutants, including ∆*braB*, ∆*ddcP(ylbL)*, ∆*ponA*, ∆*spoVG* and ∆*sftA*, for reduced FtsZ levels, by introducing an IPTG-inducible *ftsZ* allele (P*spac-ftsZ*). As a positive control we included a ∆*zapA* strain, which has been shown to be very sensitive for low cellular FtsZ levels [22], and ∆*kinE* as a negative control. The spot-dilution assay was performed on agar plates containing different IPTG concentrations. As expected, the *zapA* deletion was most sensitive (Fig 6C), whereas the *braB* mutant showed the strongest resistance to low IPTG concentrations (Fig 6C). It is not immediately clear why ∆*ponA* showed sensitivity towards lower FtsZ concentrations, whereas it resisted high 3-MBA concentrations.

## Phenotype of a ∆*braB* mutant

Deletion of *braB* resulted in a slightly lower growth rate compared to the wild type strain (Fig F in S1 Appendix), but there is no clear reason why this would cause an increased resistance to 3-MBA and to lower FtsZ levels. However, *braB* is located adjacent to *ezrA* on the genome and reads against it (Fig 7A), so the effect of a *braB* deletion on cell division might be indirect and caused by a polar effect on the activity of *ezrA*. To exclude this possibility, we inserted a xylose-inducible *braB-gfp* fusion into the ectopic *amyE* locus. The fluorescent GFP reporter was added to follow the localization of BraB in the cell. As shown in Fig 7B, expression of the fusion protein restored the sensitivity for 3-MBA in a ∆*braB* background (strain TNVS308), indicating that the BraB-GFP fusion is active, and that the cell division effect of a *braB* mutation is unrelated to *ezrA*. The BraB-GFP signal covers the cell membrane and there was no strong accumulation at division sites (Fig 7C). The slight increase in fluorescence at cell division sites can be explained by the presence of a double cell membrane at these sites, and the position of these membranes parallel to the line of sight. The fluorescence pattern suggests that BraB does not regulate cell division directly.

## Effect of ∆*braB* on the cell membrane

It is unclear how BraB, a branched-chain amino acid (leucine, isoleucine, valine) transporter, can influence cell division. *B. subtilis* contains two other branched-chain amino acid permeases, BcaP and BrnQ, and it has been shown that the absence of *braB* does not affect the uptake of branched-chain amino acids [58]. We deleted both *bcaP* and *brnQ*, but neither mutations changed the sensitivity towards 3-MBA (Fig G in S1 Appendix). While measuring the cell length of ∆*braB* cells, we did notice that the fluorescent membrane dye FM5-95 gave an irregular fluorescent membrane stain (Fig 8A). Such irregular fluorescent membrane

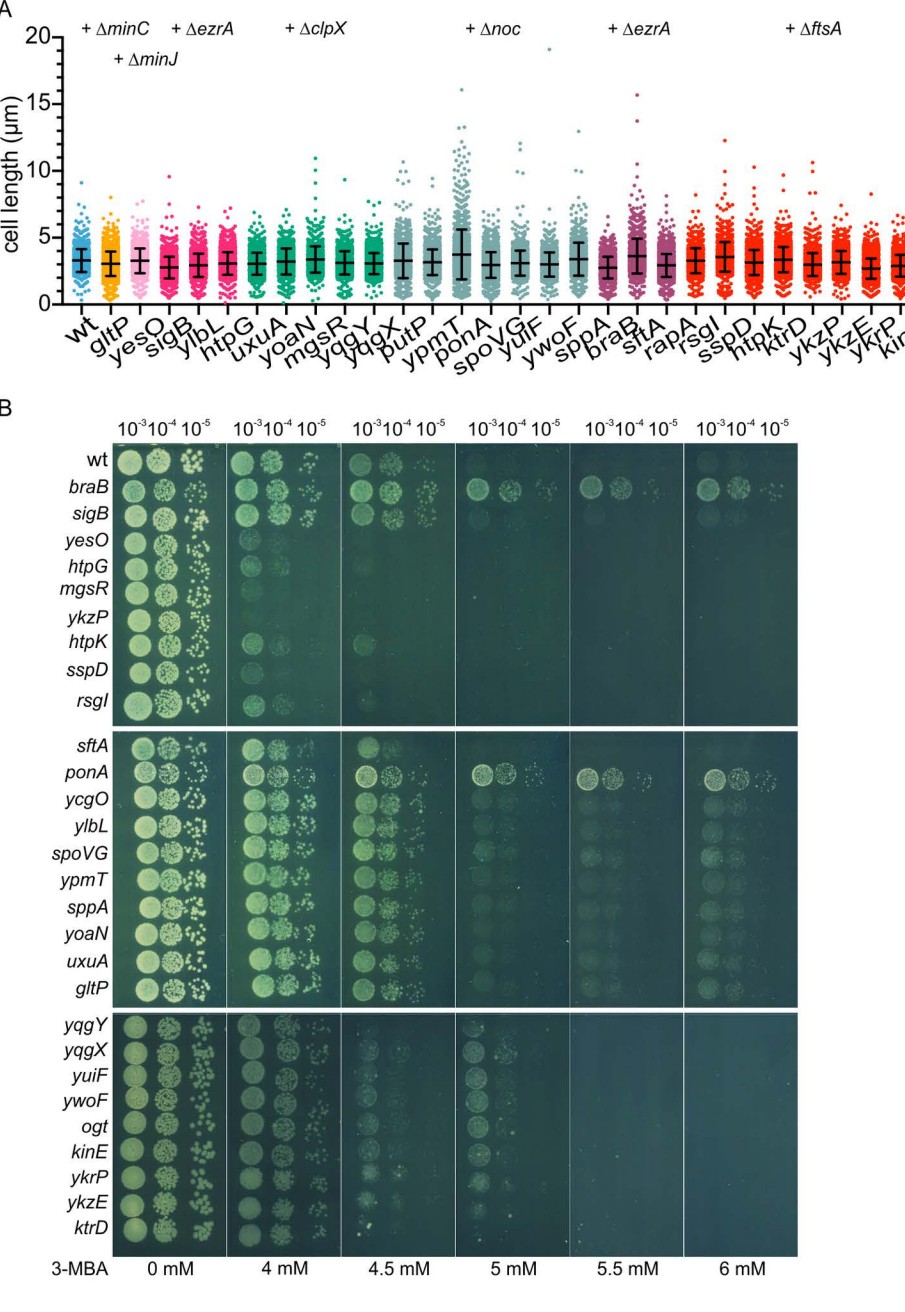

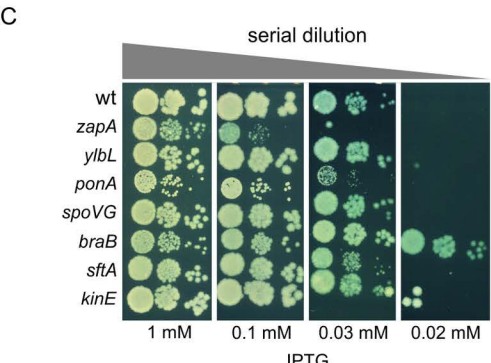

**Fig 6. Cell division characteristics of potential suppressor mutants.** (A) Cell length measurements of single deletion mutants. Colors indicate the different gene deletion step in which the related mutations appeared

(deleted gene indicated above). Based on 3 biological replicates, at least 500 cells were measured for every mutant. Mean and standard deviations are indicated. (B) Spot-dilution assay to test the sensitivity of single deletion mutants for increased concentrations of the FtsZ inhibitor 3-MBA. (C) Spot-dilution assay to test the sensitivity of single deletion mutants for decreasing FtsZ concentrations. The mutants contained an IPTG-inducible *ftsZ* allele, and were spotted on plates with decreasing concentrations of IPTG. Δ*zapA* was included as positive control and wild type and a Δ*kinE* mutant as negative controls.

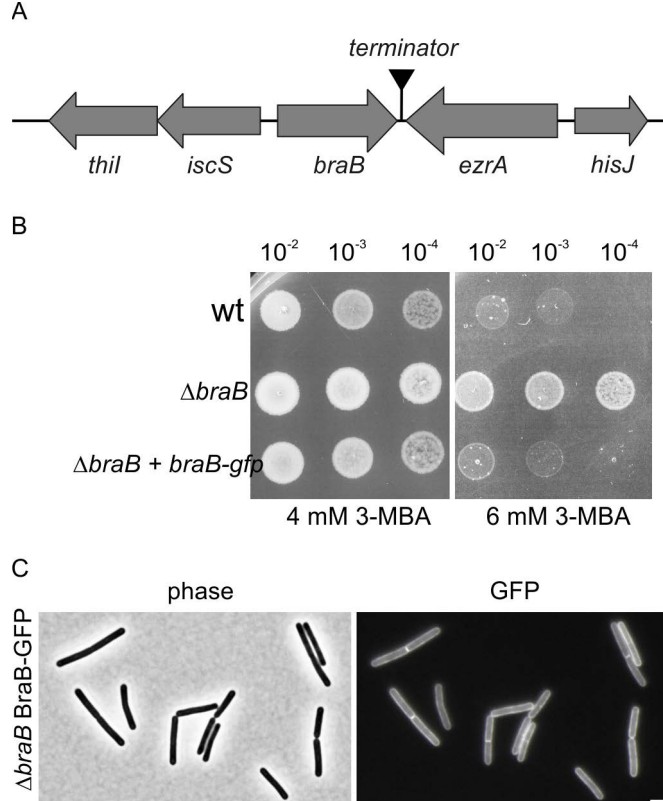

**Fig 7. BraB-GFP localization.** (A) Schematic presentation of the *braB* genome location. (B) 3-MBA spot-dilution assay, showing that the sensitivity of a Δ*braB* strain (strain TNVS292) for the FtsZ inhibitor 3-MBA is restored by ectopic expression of a BraB-GFP fusion (strain TNVS308). (C) Fluorescence light microscopy showing membrane localization of BraB-GFP in Δ*braB* (strain TNVS308). All cells were grown in LB supplemented with 0.1% xylose at 37 °C. Samples were taken at exponential phase for microscopic analyses and the spot-dilution assay. Scale bar, 2 μm.

staining has been documented before and is presumed to be caused by local enrichment of more fluid lipids, i.e., lipids containing short, branched and/or unsaturated fatty acids. These domains can increase the concentration and/or fluorescence intensity of membrane probes and are also referred to as RIFs (regions of increased fluidity) [74]. Many bacteria contain two types of branched-chain fatty acids, iso- and anteiso-fatty acids, which differ in the position of the terminal methyl side chain. The iso form, with a methyl side chain at the 2nd terminal C atom, requires final attachment of either valine or leucine to the carbon chain, whereas the anteiso form, with a methyl side chain at the 3rd terminal C atom, requires isoleucine as final attachment to the carbon chain. Thus, inactivation of BraB might influence the concentration of these fatty acids. However, a fatty acid analysis of wild type and Δ*braB* cells did not show lower levels of branched-chain fatty acids in the mutant, although there was a clear shift from anteiso (fluid) to iso (less fluid) fatty acids (Fig 8B). On the other hand, the level

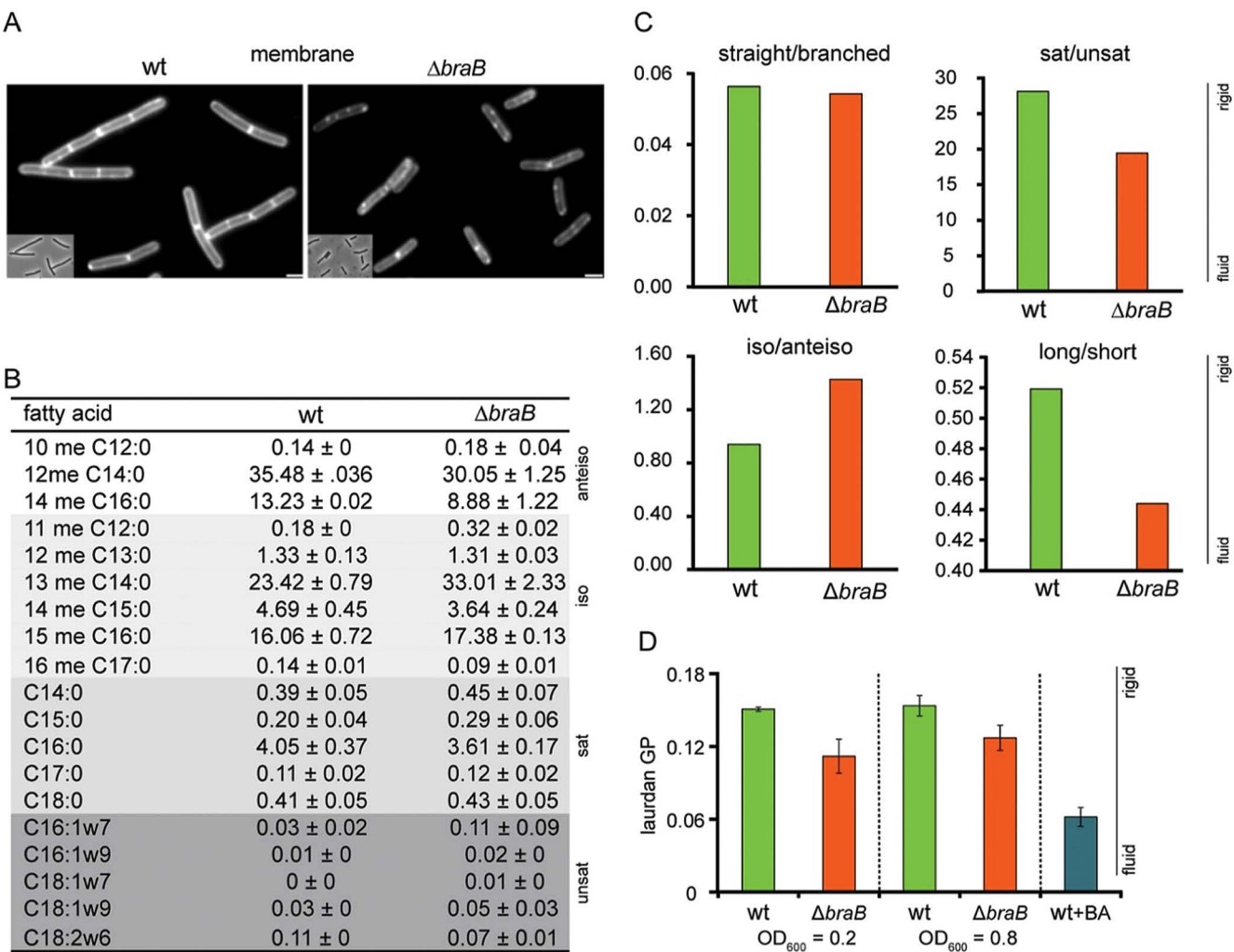

**Fig 8. *braB* deletion affects membrane composition.** (A) Aberrant fluorescent membrane pattern in exponentially growing Δ*braB* cells compared to wild type cells. Cells were stained with the membrane dye FM5-95. Wide field images with more cells are shown in Fig F in S1 Appendix. Strains used: wt and TNVS292. Scale bars, 2 μm. (B) Ratio of the fatty acid composition of the membrane are presented as mean and standard deviation of duplicate measurements. Anteiso-, iso-, saturated and unsaturated fatty acids are indicated and color-coded white to dark grey for clarity, respectively. Long fatty acids contain 16 or more carbons and short fatty acids have less. Strains used: wt and TNVS292. (C) Ratios of straight (saturated and unsaturated) and branched (iso- and anteiso), saturated and unsaturated, iso and anteiso, and long and short chain fatty acids in wild type and Δ*braB* cells based on the data in table (B). Samples were taken at an OD$_{600}$ of 0.5. (D) Membrane fluidity of wild type and Δ*braB* cells measured with Laurdan fluorescence general polarization. Samples were taken at OD$_{600}$ of 0.2 and 0.8. As a control wild type cells were treated with the membrane fluidizer benzyl alcohol (BA). The laurdan analysis was performed with 3 biological replicates.

of unsaturated and short-chain fatty acids, both more fluid, increased as well (Fig 8B). To determine whether these alterations affected the overall fluidity of the cell membrane, we performed a Laurdan fluorescence general polarization measurement. The fluorescence emission spectrum of Laurdan depends on lipid head group spreading and fatty acyl chain flexibility, making it a good indicator of membrane fluidity [75]. As shown in Fig 8C, the absence of BraB increased the overall membrane fluidity considerably.

## Discussion

We have succeeded in constructing a *B. subtilis* strain that lacks 8 conserved cell division proteins without substantially impairing growth. Six of these interact directly with FtsZ (ZapA,

EzrA, FtsA, MinC, UgtP and ClpX) and 2 play critical roles in its regulation (MinJ and Noc) (Fig 1). Clearly the emergence of suppressor mutations enabled this feat, but also the order of deletion steps has been critical. In this respect it is interesting to remark that recently it was shown that inactivation of *spX* suppresses the synthetic sick phenotype of a *minD noc* double knockout [76]. So, by deleting *spX*, with the purpose to maintain genetic competence when *cplX* would be deleted, we might have inadvertently introduced a suppressor mutant that enabled us to remove so many cell division genes.

## SepF and FtsA

Our data suggests that SepF is sufficient for the formation of an active Z-ring. SepF is conserved in Gram-positive bacteria and cyanobacteria and absent from other Gram-negative bacteria, where FtsA functions as the membrane anchor for FtsZ [12,13]. Many Gram-positives use both FtsA and SepF, like the Firmicutes, whereas, e.g., Corynebacteria only use SepF. Interestingly, archaea contain only SepF homologues and it was speculated that FtsZ and SepF may date back to the Last Universal Common Ancestor (LUCA) [15].

Cryo-EM of *Caulobacter crescentus* and *E. coli* has shown that FtsZ polymers are separated from the cell membrane by 15 to 16 nm [27,28]. This is a substantial distance that is caused by the presence of the 45 kDa membrane anchor FtsA binding to the flexible C-terminus of FtsZ [27]. It seems unlikely that late cell division proteins, which are all transmembrane proteins without large cytoplasmic domains, are able to bridge this distance and directly interact with FtsZ. In fact, for *E. coli* it has been shown that the recruitment of late cell division proteins relies on the interaction of FtsA with the cytoplasmic domain of the late cell division protein FtsN [77]. Gram-positive bacteria do not contain an FtsN homologue and it is unknown how the late cell division proteins are recruited to the Z-ring in these organisms. Our multiple deletion strain suggests that SepF, interfacing the Z-ring and cell membrane, is the link between early and late cell division proteins in Gram-positive bacteria. However, it is then likely that, at least in *B. subtilis*, this function is shared with FtsA, since a mutant lacking SepF can still divide [14].

SepF spontaneously forms strongly curved polymers with a diameter of approximately 43 nm, which is comparable to the thickness of the septal wall. In a previous study, we have shown that the specific curvature of SepF polymers determines the thickness of the division septum, likely by forming arc-like polymers that follow the leading edge of the nascent septal wall, creating precise septum wide scaffolds for FtsZ polymers and the associated proteins responsible for synthesis of the septal wall [78]. In the presence of ATP and a lipid bilayer *E. coli* FtsA can also form small ring-like polymers with a radius of about 20 nm [79]. However, there is no indication that this characteristic would somehow guide the width of the newly formed septal wall. This scaffold function of SepF might be the reason why we were unable to construct a minimal divisome mutant that uses only FtsA as membrane anchor for FtsZ.

## Branched-chain amino acid transporter BraB

Whole genome sequencing revealed a number of spontaneous suppressor mutations among which one maps to *braB*. The S343P mutation in BraB is located in a putative transmembrane domain, likely disabling the protein. A *braB* deletion does not affect the uptake of branched-chain amino acids [58], and inactivation of the other branched chain amino acid encoding transporters, *bcaP* and *brnQ*, did not convert resistance to 3-MBA. It is unlikely that the protein interacts directly with one of the structural cell division proteins since a BraB-GFP fusion showed no preference for cell division sites. In many bacteria branched chain amino acids are incorporated at the end of fatty acids and are way to regulated membrane fluidity [80].

## Role of lipids in cell division

It has long been speculated that membrane lipids play a role in bacterial cell division. More than 50 years ago Francois Jacob and co-workers showed that membrane synthesis occurs primarily in the middle of *B. subtilis* cells, and postulated that there must be an intricate relationship between DNA replication, cell division and membrane synthesis in bacteria [81]. In fact, different studies have shown increased lipid synthesis during cell division in *Bacillus* [82,83]. However, so far there is no indication for a regulatory link between lipid synthesis and cell division. Several years ago, a microscopy study suggested that the acyltransferase PlsX, which catalyzes an essential step in lipid synthesis, binds to the Z-ring [84], but this turned out to be an artefact [85,86]. Nevertheless, certain lipid species can affect cell morphology in bacteria. For example, a reduction in anionic phospholipids in *E. coli* resulted in filamentous cells [87,88]. It was shown that the membrane association of *E. coli* MinD, which recruits MinC to the cell membrane, is stimulated by anionic phospholipids but also by an increase in unsaturated acyl chains [89]. The latter observation was attributed to facilitated insertion of the C-terminal amphipathic helix of MinD, which functions as a membrane anchor. Insertion of such a relatively bulky helix in between phospholipid molecules will cost less energy when lipids are less densely packed and the membrane is in a more fluid state [90]. Since the membrane fluidity in a Δ*braB* mutant is increased it might be that this stimulates the attachment of FtsA and SepF to the cell membrane, as both proteins use an amphipathic helix as membrane anchor [14,17]. We have examined whether changing the membrane fluidity of *B. subtilis* affects its sensitivity to 3-MBA and depletion of FtsZ. Changing the membrane fluidity can be achieved by using a *B. subtilis* strain lacking the phospholipid desaturase gene *des* and the *bkd* operon, encoding the enzymes catalyzing the conversion of branched chain amino acids into intermediates for branched chain amino acid synthesis. This mutant can grow in minimal medium when supplied with either the iso- or anteiso-branched chain fatty acid precursors isobutyrate or 2-methyl butyrate, respectively. Feeding this mutant with the latter precursor will increase the membrane fluidity [91]. However, the presence of 2-methyl butyrate did not increase the resistance to either 3-MBA or depletion of FtsZ compared to Δ*des* Δ*bkd* mutant cells fed with isobutyrate (Fig H in S1 Appendix). Therefore, it seems unlikely that only a change a membrane fluidity accounts for the observed Δ*braB* phenotype.

## Minimal divisome requirements

Our research was initiated by the question what minimal set of cell division proteins are needed for division of a bacterium that contains a cell wall. Clearly FtsZ, but this is not the case for cell wall-lacking bacteria. In their pioneering work, Hutchison and coworkers constructed a minimal cell by taking a relatively simple cell wall-lacking *Mycoplasma mycoides* cell and removing all non-essential genes [3]. Interestingly, this minimal cell did not require FtsZ, although its presence was shown to improve cell division and reduce cell size [92]. Some *Mycoplasma* species lack a *ftsZ* gene by nature, and it was already shown that the *ftsZ* gene of *Mycoplasma genitalium* can be inactivated without affecting viability [4]. Presumably, FtsZ is not vital for these bacteria because they lack a cell wall. In fact, *B. subtilis* and *E. coli* mutants that grow without a cell wall, so called L-forms, can propagate without FtsZ and produce daughter cells by spontaneous membrane blebbing [5,6]. There has been one example of a cell wall-forming bacterium that can propagate when its *ftsZ* gene is inactivated and that is the filamentous actinomycetes *Streptomyces coelicolor*. This bacterium grows by branching and primarily uses cell division for dividing hyphae into multiple sporulating cells. Although the latter was blocked in the Δ*ftsZ* mutant, cells were still able to form branching hyphae [93].

When designing a cell wall-containing synthetic bacterial cell, a mechanism must be in place to divide its cell wall, and our data suggests that FtsZ and SepF are sufficient for the positioning of proteins responsible for this, although we cannot exclude that some unknown protein might have become essential for cell division in the minimal divisome mutant. Interestingly, the minimal divisome mutant BMD27 resembles the situation found in most Actinobacteria (Streptomycetes, Mycobacteria, Corynebacteria etc), which all lack MinJ, ZapA, EzrA, FtsA and the Min and Noc systems, but which do contain SepF and FtsZ [94]. It is not yet clear how Actinobacteria control correct Z-ring placement at midcell [95], however even in BMD27 most divisions still occur in between nucleoids. Possibly the density of DNA polymers, maybe enriched close to the cell membrane by transertion (coupling of transcription, translation and secretion/membrane insertion), is sufficient to prevent efficient FtsZ polymerization over the nucleoids [47,96].

Whether SepF and FtsZ are sufficient for cell division in a synthetic cell remains to be seen. Due to the absence of MinC, the BMD27 strain showed aberrant polar divisions resulting in multiple minicells, which is an undesirable trait for a synthetic cell. Actinobacteria do not contain a Min system, but the reason why they have no need for such control is unclear. These bacteria grow by cell wall synthesis at their poles, whereas Bacilli synthesizes new cell wall material along their entire cell length. Maybe this growth mode is more prone to the formation of extra cell division sites next to nascent septa, necessitating the need for a regulatory Min system. *E. coli* and related Gram-negatives follow a comparable cell wall growth modus as *Bacillus*, and also use the MinCD system to prevent formation of minicells. Of course, it cannot be ruled out that Actinobacteria use other, yet unknown, systems to prevent aberrant cell division. E.g. Streptomyces use the protein couple SsgAB to assure proper localization of Z-ring formation [97]. Thus, the choice of cell wall growth might determine whether the Min regulatory system should be included in the blueprint of a functional minimal synthetic bacterial cell.

## Conclusion

In this study we tried to define the minimal requirements for an active Z-ring in a walled bacterium. The next step is to determine the minimal set of late cell division proteins required for the actual synthesis of the septal wall. Very likely, this set will include the peptidoglycan synthesizing glycosyltransferase FtsW, transpeptidase Pbp2B, and the bitopic membrane proteins FtsL and DivIC, required for stable recruitment of late cell division proteins. These proteins are essential in *B. subtilis*, and their homologs in *E. coli* as well [26]. They are present in almost all cell wall containing bacteria including Actinobacteria. Thus, in theory, a minimal cell wall-containing Gram-positive bacterium should be able to perform septum synthesis using only 6 proteins: FtsZ, SepF, FtsW, Pbp2B, FtsL and DivIC. It will be interesting to see whether such a set can function as an autonomous 'cell division biobrick' for the construction of synthetic bacterial cells in the future.

## Materials and methods

### General methods

Strains and primers used in this study are listed in Tables A and B in S1 Appendix, respectively. *B. subtilis* strains were grown in LB medium at 37 °C and 200 RPM shaking. LB medium was supplemented with 10 mM $MgSO_4$ and 1% glucose in the case of the BMD strains. When required, erythromycin (1 μg/ml), spectinomycin (100 μg/ml), phleomycin (2 μg/ml) or tetracycline (10 μg/ml) were added. Chromosomal DNA for transformation to *B. subtilis* was prepared as described before [98]. Competent *B. subtilis* strains were transformed with either genomic DNA or PCR products using an optimized method described in [99].

## Construction of mutants

The minimal divisome mutants were constructed using a marker-free deletion method described by Morimoto et al. [43]. Detailed construction of these and the other mutants is outlined in S1 Appendix.

## Growth curves and cell length measurements

Growth curves were measured in microtiter plates, starting with overnight cultures from fresh single colonies, which were diluted to an optical density $OD_{600}$ of 0.1, grown to mid-exponential phase at 37 °C, and diluted to an optical density $OD_{600}$ of 0.05 in 150 µl.

For cell length measurements, overnight cultures were diluted to an $OD_{600}$ of 0.1 and at mid-exponential phase samples were taken, and cell membranes were fluorescently stained with 1 µg/ml FM5-95 after which cells were prepared for fluorescence microscopy analysis.

## Fluorescence microscopy

Cell membranes were stained with FM5-95 (1 µg/ml final concentration, Invitrogen) and DNA was stained with DAPI (5 µg/ml final concentration, SIGMA). Cells were immobilized on 1.3% agarose-covered microscope slides. Epifluorescence images were acquired using Nikon Ti-E and the imaging program Metamorph 6 (Molecular Devices). For SIM microscopy, cell membranes were stained with Nile red (200 ng/ml, Invitrogen). To reduce background caused by excess membrane dye, the cover glass was coated with dopamine, by incubating the glass in 2 mg/ml L-dopamine in Tris-HCl pH 7.5 buffer for at least 30 min [100]. Excess dopamine was removed by washing with demi water. 1% glucose was replaced by 1% fructose when FtsZ-GFP expression was induced with xylose (1%), because of catabolite repression of the xylose-inducible promoter [101]. SIM images were taken with a Nikon Ti-E microscope and the imaging program Nis Elements AR version 4.50.

## Whole genome sequencing

Chromosomal DNA was isolated as described in [102], and genomic DNA libraries were generated according to the manufacturers' protocols using the Ion Xpress Plus gDNA Fragment Library Preparations (Life Technologies). Bar-coded libraries were prepared using the Ion Plus fragment library kit (Life Technologies) and the Ion Xpress DNA bar coding kit (Life Technologies) according to the 200-base-read Ion Proton libraries instructions of the manufacturer. Sequencing was performed on the Ion Proton system using the Ion PI Chips (Life Technologies) according to the manufacturers' protocols. The FASTQ files were subjected to a quality control procedure, using in-house software and fastqc (www.bioinformatics.babraham.ac.uk/projects/fastqc/). Sequencing reads were mapped onto the *B. subtilis* reference genome (gi|255767013|ref|NC_000964.3|) using Tmap [103]. Finally, single nucleotide variants, insertions and deletions were identified using the flow-aware Torrent Variant Caller (TVC). From these variants, insertions and deletions mutations with 75% frequency and coverage of 80 were scored as relevant.

## Spot-dilution assays

To test the sensitivity for 3-MBA, single colonies were cultured in LB to an $OD_{600}$ of 0.5 and serial dilutions were prepared in prewarmed LB at 37 °C. The dilutions were spotted onto LB agar plates with appropriate concentration of 3-MBA. When an inducer was required, the appropriate concentration was included in the LB agar plates. The plates were incubated overnight at 37 °C.

FtsZ-depletion sensitivity assays made use of an IPTG-inducible *ftsZ* gene (*Pspac-ftsZ*) [31]. Strains were streaked on LB plates containing 200 µM IPTG and 2 µg/ml phleomycin and grown at 37 °C. Single colonies were grown to exponential phase in LB supplemented with 200 µM of IPTG and 2 µg/ml phleomycin, then 10-fold serial dilutions were prepared in fresh pre-warmed LB. 10 µl of appropriate serial dilutions were spotted on LB plates with 2 µg/ml phleomycin and different concentration of IPTG. The presence of phleomycin was necessary to prevent excision of the *Pspac-ftsZ* construct, which was integrated by single crossover.

### Fatty acid analyses

For fatty acid composition analysis, 500 ml cultures of *B. subtilis* 168 and TNVS292 (Δ*braB*) were grown at 37 °C in LB until an $OD_{600}$ of 0.5. Cultures were quickly cooled on slush ice followed by centrifugation at 12,000 x g. Cells were washed with 2 ml ice-cold 100 mM NaCl and dry cell pellets were flash-frozen in liquid nitrogen. Samples were lyophilized, covered with argon, and stored at −80 °C. Lyophilized cell pellets were dissolved in 1.5 ml PBS and disrupted by ultra-sonication. 50 µl of the cell lysate were prepared for gas chromatography as described [104]. In short, samples were mixed with 10 nmol internal fatty acid standard and transmethylated in 1 ml 3 M HCl at 90 °C for 4 h, followed by hexane extraction. Extracts were dried under argon stream, resuspended in 100 µl hexane, and subsequently injected into a Hewlett Packard GC 5890 gas chromatograph equipped with an Agilent J&W HP-FFAP 25 m, 0.20 mm, 0.33 µm GC column. Fatty acid methyl esters were detected by flame ionization.

### Membrane fluidity measurements

To assess the membrane fluidity of *B. subtilis*, the fluorescent membrane probe Laurdan was used [75]. Briefly, *B. subtilis* 168 and TNVS292 (Δ*braB*) were grown at 37 °C in LB until an $OD_{600}$ of 0.2 and 0.8 and incubated with 10 µM Laurdan (from a 1 mM stock in DMF) for 5 min. Cells were then quickly washed four times with pre-warmed PBS containing 0.2% glucose and 1% DMF. Cell pellets were resuspended in the same buffer and OD was adjusted to 0.4 prior to transferring 150 µl aliquots to a pre-warmed black microtiter plate. Laurdan fluorescence was excited at 350 nm, and emission at 460 and 500 nm was recorded. Laurdan GP was calculated according to the formula (I460-I500)/(I460+I500).

### Supporting information

**S1 Appendix. Strain construction and supplementary figures.** This file describes the construction of BMD and other mutants listed in Table A (*B. subtilis* strains used in this study), and the primers used for the construction, listed in Table B (Primers used in this study). It includes the following supplementary figures: Fig A: Gene deletion pathways, Fig B: Growth of Bacillus minimal divisome strains, Fig C: FtsZ and nucleoid localization in BMD27, Fig D: FtsZ and nucleoid localization in *ftsA::erm*, Fig E: BMD27 in stationary phase, Fig F: Phenotype of the *braB* deletion mutant, Fig G: Sensitivity of Δ*brnQ* and Δ*bcaP* mutants for 3-MBA, Fig H: Effect of membrane fluidity on sensitivity for FtsZ perturbations, and a list of References.
(PDF)

**S1 Raw Data. Raw data for Fig 2C.**
(CSV)

**S2 Raw Data. Raw data for Fig 4.**
(CSV)

**S3 Raw Data.** Raw data for Fig 6A.
(CSV)

## Acknowledgments

We would like to thank all members of the lab for constructive and critical discussions.

## Author contributions

**Conceptualization:** Leendert W. Hamoen.

**Data curation:** Terrens N. V. Saaki, Michaela Wenzel, Simon Syvertsson, Tjalling K. Siersma.

**Formal analysis:** Ilkay Celik Gulsoy, Terrens N. V. Saaki, Michaela Wenzel, Simon Syvertsson, Tjalling K. Siersma, Leendert W. Hamoen.

**Funding acquisition:** Leendert W. Hamoen.

**Investigation:** Ilkay Celik Gulsoy, Terrens N. V. Saaki, Simon Syvertsson, Taku Morimoto, Tjalling K. Siersma.

**Methodology:** Taku Morimoto.

**Project administration:** Leendert W. Hamoen.

**Resources:** Taku Morimoto.

**Supervision:** Terrens N. V. Saaki, Leendert W. Hamoen.

**Validation:** Ilkay Celik Gulsoy, Terrens N. V. Saaki, Simon Syvertsson.

**Visualization:** Ilkay Celik Gulsoy, Terrens N. V. Saaki, Michaela Wenzel, Tjalling K. Siersma.

**Writing – original draft:** Ilkay Celik Gulsoy, Terrens N. V. Saaki.

**Writing – review & editing:** Leendert W. Hamoen.

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
