## [Decision Letter · Decision Letter 0]

12 Sep 2024

Dear Dr Hamoen,

Thank you very much for submitting your Research Article entitled 'Minimization of the Bacillus subtilis divisome suggests FtsZ and SepF can form an active Z-ring, and reveals the amino acid transporter BraB as a new cell division influencing factor' to PLOS Genetics.

The manuscript was fully evaluated at the editorial level and by independent peer reviewers. The reviewers appreciated the attention to an important topic but identified some concerns that we ask you address in a revised manuscript.

We therefore ask you to modify the manuscript according to the review recommendations. Your revisions should address the specific points made by each reviewer. It is also important to address the concern of Reviewer #3 that there is data that is "data not shown." Please be aware that PLOS Genetics requires all data to be shown, otherwise references to such data must be removed.

2) Upload a Striking Image with a corresponding caption to accompany your manuscript if one is available (either a new image or an existing one from within your manuscript). If this image is judged to be suitable, it may be featured on our website. Images should ideally be high resolution, eye-catching, single panel square images. For examples, please browse our archive . If your image is from someone other than yourself, please ensure that the artist has read and agreed to the terms and conditions of the Creative Commons Attribution License. Note: we cannot publish copyrighted images.

If present, accompanying reviewer attachments should be included with this email; please notify the journal office if any appear to be missing. They will also be available for download from the link below. You can use this link to log into the system when you are ready to submit a revised version, having first consulted our Submission Checklist .

PLOS has incorporated Similarity Check , powered by iThenticate, into its journal-wide submission system in order to screen submitted content for originality before publication. Each PLOS journal undertakes screening on a proportion of submitted articles. You will be contacted if needed following the screening process.

To resubmit, log into your Editorial Manager account and select the option 'Revise Submission' in the 'Submissions Needing Revision' folder.

Yours sincerely,

Danielle A. Garsin

Academic Editor

PLOS Genetics

Sean Crosson

Section Editor

PLOS Genetics

Reviewer's Responses to Questions

**Comments to the Authors:**

Reviewer #1: This interesting paper by Gulsoy et al. seeks to identify the minimal set of genes required for cell division of a Gram-positive walled bacterium (the model Bacillus subtilis). In E. coli and B. subtilis model systems, cell division consists of a setup stage in which FtsZ polymers form a midcell Z ring by attaching to the cytoplasmic membrane via adaptor proteins, a recruitment stage wherein the septum-synthesis enzymes are brought to the Z ring, followed by a septum synthesis stage. In this study, the authors tested the minimal gene requirements for formation of the first stage--an active cytokinetic Z ring—by sequential deletion of conserved genes involved in cell division, starting with the least crucial and ending with the more crucial genes. After trying out several schemes for sequentially deleting the genes, they found that most of the conserved early stage division genes could be deleted, including ftsA and ezrA, and cells left with only sepF and ftsZ in this first stage were able to divide, albeit poorly. This reliance mainly on SepF and FtsZ is perhaps not too surprising, as Gram-positive Actinobacteria carry only SepF and FtsZ as their essential cell division genes (mentioned on lines 453-456) and not the others. So, the authors basically forced B. subtilis to become Actinobacteria in terms of cell division, with Actinobacteria being closer to a minimal cell than B. subtilis.

Many of the gene deletions nevertheless gave rise to suppressors, some of which were known to regulate cell division. The authors did a nice job in carefully identifying the suppressors and testing their effects, although most of them did not lead to any obvious reason for why they arose in response to the gene deletions. There was one suppressor with strong effects on cell division, however: a loss of function allele in braB, which encodes a branched chain amino acid transporter. A deletion of braB (which does not affect the uptake of branched-chain amino acids) was notable in that it resisted the effects of an anti-FtsZ compound, 3-MBA, and allowed normal cell division to occur at artificially low levels of FtsZ. These traits, along with the ability of the braB knockout mutant to divide at shorter cell lengths than WT cells, and the lack of BraB-GFP localization at division septa, suggested that the activity of BraB somehow acts as an indirect inhibitor of cell division. Notably, cell membranes in a braB mutant had different lipid composition and higher overall fluidity, including visible membrane domains of higher fluidity. It is suggested that this higher membrane fluidity in the absence of BraB somehow promotes cell division. Although it is exciting to have found a new factor in promoting cell division in B. subtilis, it was also frustrating that so little was gleaned from the suppressor findings, especially as the links between branched chain amino acid transport and membrane fluidity are tenuous. Otherwise, the paper is well written and clear, and provides some new insight into requirements for cell division proteins in B. subtilis. The Discussion is thorough and thoughtful.

Specific comments:

Lines 79-80: There are a few exceptions to this. Chlamydia, which has a cell wall, uses MreB to divide instead of FtsZ. And Planctomycetes lack FtsZ.

Line 92: replace “also displays some inhibitors activity when it comes to” with “can inhibit”

Line 102: should be “proteins” in both instances.

Line 105: replace “can amount to” with “is approximately”

Line 114: replace “couple” with “pair”

Line 148: replace “in” with “map to”

Line 149: should be “led”

Line 207: Do the authors have an explanation for why multiple minicells form at the expense of division septa elsewhere? Is there some kind of positive feedback occurring?

Line 221: E. coli lacking Min and SlmA also still show a preference for Z rings between nucleoids, so this may be a conserved trait (PMID: 25101671).

Line 301 (and 1085) should be “IPTG”

Lines 306-307: lower FtsZ concentrations should have a different effect from high 3-MBA concentrations, as the latter should stop FtsZ polymer treadmilling, unlike the former. Perhaps FtsZ-mediated septum formation in B. subtilis (which PonA contributes to) is somewhat treadmilling-independent, while it absolutely requires a minimum number of FtsZ molecules?

Lines 319-320 and Fig. 7C: How do the authors know for sure that there was no accumulation of BraB-GFP at division sites? Were the levels of fluorescence at septal membranes compared with those of a bona fide division protein or a known non-division membrane protein?

Line 373-375: although FtsA-FtsN interaction is important for FtsN’s recruitment and subsequent activation of the septum synthesis enzymes, other key division proteins such as FtsQLB can be recruited independently of FtsN (and FtsA-FtsN interactions). Indeed, FtsN can be partially or completely bypassed with hyperfission alleles.

Line 394: replace “one in” with “one maps to”

Line 437-438: this is not a full sentence

Line 453: should be “positioning”

Line 452: are FtsZ and SepF really “sufficient”? Other unknown proteins may play a nonessential role that may become essential in the BMD27 mutant.

Line 460: Z rings also still are biased to form between nucleoids of E. coli cells that lack the nucleoid occlusion and Min systems (Bailey?)

465: synthesize

468: Some Gram negative cells such as the rhizobiales grow via polar growth as well.

Reviewer #2: The manuscript by Gulsoy et al describes the construction of a Bacillus subtilis strain lacking eight conserved cell division proteins, with the aim of identifying the minimal divisome machinery. The work indicates that FtsZ and SepF are key to form an active Z-ring and proposed a new role for BraB in cell division. This is a considerable effort, of interest to the community studying bacterial cell division.

General comments

Paragraph starting in line 177 – Authors mention that they tried different sequences of successive deletion of division genes but most resulted in a lethal phenotype. A supplementary figure with the failed paths should be included in the paper, as these can be informative regarding negative genetic interactions and also inform other researchers on paths that are not worth to follow.

Line 205 (and various places in the manuscript) – mention strain name. This helps the reader in case they need to check the genotype

Line 211 – The described results were not obtained with strain BMD27, as mentioned in the text, but with a derivative of this strain expressing FtsZ-GFP.

Line 215 – stating “approximately half of the Z-rings…” instead of “the majority of Z-rings” would be more in line with the data.

Section starting on line 226 and Fig 5 – There are various suppressor mutations in BMD strains that appear and disappear on the genome of sequentially made strains. For example, the gltP mutation is present in BMD2, absent in BMD 2,3 and 5, and present again in BMD6. Then absent in BMD 7 and 9, but present again in BMD 12, 14 and 27. How do author explain this if the strains were made sequentially? This is a key point that has to be explained.

Line 311-312 – referring to deletion of braB, authors state: “there is no clear reason why this would cause an increased resistance to 3-MBA and lower FtsZ levels”. This sentence implies “there is no clear reason why this (deletion of braB ) would cause (…) lower FtsZ levels”, which was not shown. Do authors mean “increased resistance to 3-MBA and to lower FtsZ levels”?

Line 418 - briefly explain why DivIB was not considered here.

Figure 2 – it is difficult to distinguish some of the colours. Maybe ordering colours in a gradient would help? Also, in the labels of the graphs in this figure “Dnoc” should be indicated as “Dnoc + EzrA”. Otherwise the reader maybe misled. In the legend of this figure, spx should be mentioned, so that it is clear that it is not a division protein

Figure 7 B – add strain names to the right of the images. Also in both panel B and C there is a strain labelled DBraB+braB-gfp, but in the legend two different strains are mentioned. This may be confusing to the reader. The title of this figure can be improved.

Minor comments/typos

Lines 101, 102 and 108 – replace “the late cell division protein” by “late cell division proteins”(plural) or “the late cell division protein complex”

Line 134 – Minc – “C” should be capital letter.

Line 149 – “we also found an unknown suppressor mutation” – all suppressor mutations are unknown before genome sequencing. Maybe “unexpected suppressor mutation”?

Line 200 – remove comma after BMD27

Line 213 – do authors mean dissect or bisect?

Line 974 – remove “a” before “more details”

Line 1045 – strains should not be plural

Line 1047 – “seems to bisect” instead of “seem to dissect”?

Figs S2 and S3 – Phase contrast images are too small. They could be shown to the right of the bottom fluorescence image.

Reviewer #3: This paper sets out to determine the minimal set of cell division genes that are essential. They cleverly sequentially delete as many cell division genes as possible, finding that only FtsZ and SepF are essential. This is the first study in a cell wall containing bacterium to identify this minimal set. This is the strongest part of the manuscript. The finding that loss-of-function mutations of BraB suppress the cell growth defect of the minimal FtsZ, SepF mutant are interesting, but the conclusions about how this protein may influence cell division are unclear. I detail a number of comments would strengthen the manuscript below:

1. Line 207, page 10, indicate that multiple polar minicells are observed in Figure 3B. This statement needs to be removed on better data presented. It was not possible to see these minicells in figure 3B, possibly because the membrane stain appears so weak.

2. Line 209, page 10, conclude that the BMD27 cells are only delay in cell division because the filamentous cells divided and became smaller in stationary phase. This is an important point, but the data is not shown. Please show this data in the supplemental material.

3. Lines 221-224, page 10, indicate that FtsA is required for midcell Z-ring formation, and that this is likely due to the formation of very long filamentous cells rather than a regulatory role. However, as shown in Figure 2C, the BMD14 mutant, which does form midcell Z-ring, is similarly long and filamentous as the FtsA mutant. This would argue for a more regulatory role for FtsA. Please address this confusing data in the text of the manuscript.

4. Line 319, page 14, indicated that the BraB-GFP signal does not accumulate at division sites. However, as shown in Figure 7C, there does appear to be a central focus of BraB-GFP. The presence of this focus should be address in the text of the manuscript.

5. Line 397, page 19, indicated that inactivation of other branch chain amino acid transported did not affect resistance to 3-MBA, as inactivation of BraB did. This data is not shown, but it is important enough to show in the supplemental results.

6. Line 399, page 19, states that the BraB-GFP fusion showed no preference for the cell division sites. This line should be changed as the data does not seem to support this conclusion.

7. One interesting question about BraB is whether it is the loss of its enzymatic activity that is required to suppress the loss of cell division genes. As of now, there is a correlation between changes in membrane composition and suppression of cell division defects, but this could be due to a second function of BraB that is independent of its role in forming branched-chain fatty acids. It is not clear whether there is a particular amino acid change that can be made that would disrupt it activity as a symporter, but testing such a mutant could add significantly to your model of how BraB affect cell division.

8. Line 431, page 20, indicates that you have data showing that changing the membrane fluidity, independently of BraB inactivation, did not result in cell division suppression. This data is not shown, and again this is an important point that should be shown in the figure. This evidence, which is only brought up in the discussion, further argues the need to understand whether the enzymatic activity of BraB is important.

9. On line 102, on page 5, please indicate to which late cell division proteins you are referring.

10. Line 287, page 13, indicated that strains with deletion mutations were constructed. This should indicate whether this was done in the background of a BMD mutant or the wild-type background.

11. On Figure 2B, the results are hard to see because the lines are so close to each other and the colors so similar to each other. Adding numbers for the growth rate would help make the data in this figure more interpretable.

12. Figure 3A and 3C should address why the membrane stain shows as a couple of spots and not the length of the cell.

13. Figure 8C, it is not clear from the legend how the data in figure 3C was calculated. A formula should be added for clarity.

**Have all data underlying the figures and results presented in the manuscript been provided?**

Reviewer #1: Yes

Reviewer #2: Yes

Reviewer #3: Yes

PLOS authors have the option to publish the peer review history of their article (what does this mean? ). If published, this will include your full peer review and any attached files.

**Do you want your identity to be public for this peer review?** For information about this choice, including consent withdrawal, please see our Privacy Policy .

Reviewer #1: No

Reviewer #2: No

Reviewer #3: No

---

## [Decision Letter · Decision Letter 1]

6 Jan 2025

Dear Dr Hamoen,

We are pleased to inform you that your manuscript entitled "Minimization of the Bacillus subtilis divisome suggests FtsZ and SepF can form an active Z-ring, and reveals the amino acid transporter BraB as a new cell division influencing factor" has been editorially accepted for publication in PLOS Genetics. Congratulations!

Yours sincerely,

Sean Crosson

Section Editor

PLOS Genetics

Aimée Dudley

Editor-in-Chief

PLOS Genetics

Anne Goriely

Editor-in-Chief

PLOS Genetics

Comments from the reviewers (if applicable):

Reviewer's Responses to Questions

**Comments to the Authors:**

Reviewer #1: The authors have done an excellent job in responding to my criticisms/suggestions as well as those of the other reviewers.

Reviewer #2: The authors have carefully addressed all my concerns. The new figure S1 was done in a nice, easy to grasp, way.

I have no further comments.

**Have all data underlying the figures and results presented in the manuscript been provided?**

Reviewer #1: Yes

Reviewer #2: Yes

PLOS authors have the option to publish the peer review history of their article (what does this mean? ). If published, this will include your full peer review and any attached files.

**Do you want your identity to be public for this peer review?** For information about this choice, including consent withdrawal, please see our Privacy Policy .

Reviewer #1: No

Reviewer #2: No

**Data Deposition**

http://datadryad.org/submit?journalID=pgenetics&manu=PGENETICS-D-24-00853R1

**Press Queries**

---

## [Editor Report · Acceptance letter]

PGENETICS-D-24-00853R1

Minimization of the Bacillus subtilis divisome suggests FtsZ and SepF can form an active Z-ring, and reveals the amino acid transporter BraB as a new cell division influencing factor

Dear Dr Hamoen,

We are pleased to inform you that your manuscript entitled "Minimization of the Bacillus subtilis divisome suggests FtsZ and SepF can form an active Z-ring, and reveals the amino acid transporter BraB as a new cell division influencing factor" has been formally accepted for publication in PLOS Genetics! Your manuscript is now with our production department and you will be notified of the publication date in due course.

With kind regards,

Anita Estes

PLOS Genetics

On behalf of:
